# Transience of the Retinal Output Is Determined by a Great Variety of Circuit Elements

**DOI:** 10.3390/cells11050810

**Published:** 2022-02-25

**Authors:** Alma Ganczer, Gergely Szarka, Márton Balogh, Gyula Hoffmann, Ádám Jonatán Tengölics, Garrett Kenyon, Tamás Kovács-Öller, Béla Völgyi

**Affiliations:** 1Szentágothai Research Centre, University of Pécs, H-7624 Pécs, Hungary; alma@pte.hu (A.G.); jerjely@gamma.ttk.pte (G.S.); balogh.marton@pte.hu (M.B.); hgyula@gamma.ttk.pte.hu (G.H.); jonathan@gamma.ttk.pte.hu (Á.J.T.); kovacs-oller.tamas@pte.hu (T.K.-Ö.); 2Department of Experimental Zoology and Neurobiology, University of Pécs, H-7624 Pécs, Hungary; 3MTA-PTE NAP 2 Retinal Electrical Synapses Research Group, H-7624 Pécs, Hungary; 4Center for Neuroscience, University of Pécs, H-7624 Pécs, Hungary; 5Los Alamos National Laboratory, Computer & Computational Science Division, Los Alamos, NM 87545, USA; gkenyon@lanl.gov

**Keywords:** photoreceptor, bipolar cell, amacrine cell, ganglion cell, outer plexiform layer, inner plexiform layer, ganglion cell layer, retina, parallel signaling

## Abstract

Retinal ganglion cells (RGCs) encrypt stimulus features of the visual scene in action potentials and convey them toward higher visual centers in the brain. Although there are many visual features to encode, our recent understanding is that the ~46 different functional subtypes of RGCs in the retina share this task. In this scheme, each RGC subtype establishes a separate, parallel signaling route for a specific visual feature (e.g., contrast, the direction of motion, luminosity), through which information is conveyed. The efficiency of encoding depends on several factors, including signal strength, adaptational levels, and the actual efficacy of the underlying retinal microcircuits. Upon collecting inputs across their respective receptive field, RGCs perform further analysis (e.g., summation, subtraction, weighting) before they generate the final output spike train, which itself is characterized by multiple different features, such as the number of spikes, the inter-spike intervals, response delay, and the rundown time (transience) of the response. These specific kinetic features are essential for target postsynaptic neurons in the brain in order to effectively decode and interpret signals, thereby forming visual perception. We review recent knowledge regarding circuit elements of the mammalian retina that participate in shaping RGC response transience for optimal visual signaling.

## 1. The Wiring of the Mammalian Retina

The mammalian retina is a valuable and common model in neuroscience, due to its well-defined layered structure and its excellent potential for conducting in vitro electrophysiological studies. This has allowed scientists to map many of the neuronal connections and build a wiring blueprint (Figure 1) that brings functional and morphological connections in the retina together in one schematic [1]. Elements of this blueprint are generally found in all mammalian species, including both primate and non-primate models and humans as well. However, much of the information contributing to this review originates from experimental work carried out in the mouse retina, which became a popular model due to the large number and easy access to the various genetically modified strains. Our current knowledge regarding neurons in the mouse retina points to at least 130 neuronal cell types, with 2 distinct photoreceptor types (PRs) [2], 1 type of horizontal cell (HC) [3], 15 different bipolar cell types (BCs) [4], roughly 63 diverse amacrine cell types (ACs) [5], and an estimated number of around 46 retinal ganglion cell types (RGCs) [6]. Just like in any other species, visual perception in mice begins at the level of PRs, where rods and cones create two separate pathways for daytime (high intensity or photopic) or nighttime (low intensity or scotopic) illumination, respectively. Rods are sensitive enough to detect a single photon [7], and thus they are most efficiently utilized in low-light environments. Cones, on the other hand, express photopigments that are less sensitive to the intensity of the stimulating light, but the variety in the expressed chromophores allows them to specialize to a certain wavelength, which has proven to be the first step in establishing color vision. The human retina contains 3 cone subtypes specifically adapted to peak sensitivity at short (S, ~430 nm), medium (M, ~530 nm), and long (L, ~560) wavelengths [8], while the mouse retina only possesses S and M sensitive cones [2].

PRs form synapses with 15 different types of BCs, with one BC collecting inputs from about 5–10 presynaptic PRs [9]. BCs lay the foundation for a number of parallel signaling pathways at their level [10] and the segregation of information is maintained throughout the downstream circuitry to higher visual brain centers. Roughly half of these parallel pathways inform the brain about an increase in light intensity (ON pathways), whereas the other half signals decrements in light intensity (OFF pathways). This ON and OFF polarity distinction appears to be so fundamental for vision that ON and OFF signals are established as early as the first retinal synapse (the PR-BC synapse). BCs that express ionotropic glutamate receptors (BCs: 1(ab), 2, 3(a,b), 4) are responsible for forming the OFF signaling routes [11,12], whereas those that utilize metabotropic glutamate receptors (5(a–d), 6, 7, 8/9) are responsible for conveying ON signals [11]. The first synaptic layer of the retina is also equipped with HCs that express inhibitory GABA as a transmitter and signal laterally, thereby providing feedback and feedforward inhibition to PRs and BCs, respectively [13]. This inhibition serves mainly as a spatial filter and thus forms the basis for an inhibitory surround receptive field of downstream retinal neurons and thus contrast detection for vision [14].

ON and OFF BCs both release glutamate and synapse with ON, OFF, or ON–OFF RGCs and a large variety of ACs. In turn, BCs receive AC inhibitory inputs at their axon terminals through reciprocal feedback and/or feedforward mechanisms [15]. ACs are commonly classified into glycinergic narrow or GABAergic wide-field populations [16]. ACs in general exhibit great diversity in terms of morphology and function, and according to a recent study, their molecular makeup indicates as many as 63 separate subtypes [5]. While narrow-field ACs modulate local glutamate input of BCs, wide-field ACs generally provide a non-linear surround to RGCs [17]. Through gap junction (GJ) coupling, ACs can also relay excitatory input to neighboring ACs or even to neurons of other cell types. One of the best-known examples of ACs with GJ coupling are the narrow-field AII ACs that, upon receiving excitatory glutamate input from rod BCs [18], use gap junctions to excite ON cone BCs, and glycinergic chemical synapses to inhibit OFF BCs at the same time [19]. ACs might also provide a sign-inverting feedforward mechanism, such as in the case of S-M (short wavelength–medium wavelength) color opponency in the murine retina, where an S-ON/M-OFF RGC will receive direct excitatory input from S-ON BCs and inhibition through a sign-inverting AC connection from an M-ON BC [20]. Interestingly, there is at least one AC type (VGlut3 expressing AC) that utilizes a glutamatergic excitatory signaling mechanism to provide a feedforward (ON to ON, OFF to OFF) and crossover (ON to OFF, OFF to ON) glutamatergic input in the case of direction-selective ganglion cells (DSGCs) [21]. Once visual signals have passed through all the above-mentioned circuit elements, they are integrated by RGCs to forge spike trains that inform the brain about the change in only one specific visual aspect [22].

## 2. RGC Response Transience and Possible Visual Functions

Light-evoked RGC responses have been characterized by their polarity (ON, OFF, and ON–OFF), sensitivity to various stimuli, and their kinetics. In general, the two main response parameters that are considered and studied when it comes to RGC response kinetics are the response delay and the response decay. The response delay reflects the speed by which a certain neuron reacts to an incoming stimulus, and thus neurons including RGCs are generally considered either brisk or sluggish. On the other hand, response decay reveals the ability of a neuron to maintain its activity without inactivation in response to continuous stimulation. Based on different manifestations of the latter phenomenon, neurons are known to display either maintained spiking activity (sustained response) or present a brisk spike burst (transient response) in response to continuous stimulation (Figure 2). Both aspects (response delay and decay) likely contribute to signal efficiency on postsynaptic neuronal targets in higher visual centers [23,24,25]. The transient/sustained dichotomy of RGC response decay has been documented in a variety of vertebrate species, including cold-blooded animals, primates, and non-primate mammals as well [26,27,28,29,30,31,32,33,34,35,36], indicating that this kinetic feature has been conserved throughout vertebrate evolution and thus further attesting the importance of the role that RGC response kinetics play in visual coding. However, regardless of our current understanding of the light response temporal characteristics, there are only few comprehensive models describing the exact origins and functions of transient and sustained responses [10]. Our knowledge of how RGC light response kinetics contribute to the visual signal serves basic neuroscience and can be utilized to refine robotic vision and pave the road for novel advancements in the development of retinal prosthetics, a goal that becomes markedly important by the day. Therefore, in this present review, we will focus on the diversity of RGC response decay (or transience). In addition, we also discuss how elements in the retinal circuitry participate in shaping RGC response transience to encode visual information.

## 3. Circuit Elements That Contribute to Response Transience

### 3.1. The Photoreceptor to Bipolar Cell Synapses in the Outer Retina

Since all photoreceptors generate sustained responses upon illumination [37], a sustained-to-transient response transformation must occur along the vertical axis of the retinal circuitry. At the first synaptic site in the outer plexiform layer (OPL) of the retina, rods contact only a single subtype of BCs, known as the rod BC, whereas cones distribute information to 14 cone BC subtypes that can be distinguished using both morphological and physiological measures [4,9,10,38,39]. In the outer retina, the above-mentioned sustained-to-transient response transformation could potentially be performed by quick desensitization of postsynaptic BC glutamate receptors or through an inhibitory mechanism localized to the OPL.

#### 3.1.1. Postsynaptic BC Glutamate Receptors

It was suggested over four decades ago that aspartate and glutamate are released by retinal neurons, mediating either sustained or transient signals for postsynaptic neurons [40]. Later on, converging evidence showed that the distinction between decay types is more likely to be established by the postsynaptic receptors instead. Previous work in the salamander and rabbit retinas theorized that response transience is determined by the kinetics of the postsynaptic glutamate receptors (Figure 3; mGluR_6_, AMPA, kainate) at the site of the very first retinal contact, the PR-to-BC synapse [41,42,43,44]. In this scheme, OFF BCs that are in direct contact with the photoreceptors may perform a signal transition from sustained to transient when they express AMPA receptors, but maintain sustained responses when the signal is conveyed through kainate receptors (Figure 3). This hypothesis is based on results showing that AMPA receptors display rapid recovery from desensitization, thereby transmitting high temporal frequency signals, whereas kainate receptors encode lower temporal frequencies [42]. Some data seem to agree with this hypothesis and demonstrate the proposed correlation between OFF BC responses and expressed postsynaptic glutamate receptors [42,43,45]. Other observations, however, argue that the situation is more complex and other factors, such as variations in both the AMPA and kainate subunit compositions, are relevant. In this scheme, both AMPA- and kainate-mediated OFF BC signals can be toggled towards either the transient or the sustained end of the scale in a subunit-dependent manner. In the primate retina, for example, the diffuse BCs (DB2 and DB3b) were shown to express the GluK1 kainate subunit and displayed transient responses, whereas the flat midget BCs (DB1 and DB3a) that lacked the GluK1 subunit produced sustained responses [12]. A subunit selectivity has been detected for ground squirrel OFF BCs as well, where GluK1 and GluK5 comprised the kainate receptors of Cb3a/b cells, GluK1 was present in Cb1a/b cells, and AMPA receptor subunits were expressed by Cb2 cells [46]. However, both primate and mouse retinal OFF BCs seem to utilize mainly (if not exclusively) kainate receptor-mediated glutamate signaling to generate both transient and sustained OFF BC responses [12,43,47,48]. In this respect, there is some discrepancy in the relevant literature as, besides kainate-mediated signaling of type 2 and type 3a BCs in the mouse retina, type 1 BCs seem to exclusively utilize AMPA receptors, and types 3b and 4 display a combination of AMPA and kainate receptors [49,50,51]. Nevertheless, it seems that the dominant receptor component is the GluK1 kainate subunit, and the variety in the temporal response properties results from the selective association and combination with others, including GluK3, GluK5, and AMPA channel-forming units, as well as with association proteins such as Neto1 [12,46]. 

In contrast to the OFF signaling routes, information through the ON polarity pathway is passed from both rod and cone photoreceptors to BCs via a single glutamate receptor type, the mGluR_6_ receptor (Figure 3). Contrary to this significant difference in signaling, both sustained and transient ON BCs exist simultaneously in most examined mammalian species [10]. Moreover, ON RGC responses display a response transience distribution pattern akin to those of their OFF counterparts [52] (Figure 3). Is it possible, then, that response characteristics of ON BC light responses depend on the intracellular molecular milieu of the mGluR_6_ receptors or other yet unrevealed factors? As a matter of fact, Awatramani and Slaughter [41] found that transient and sustained ON BC responses in the salamander retina differed in their sensitivity to metabotropic glutamate receptor antagonist CPPG (RS-α-cyclopropyl-4-phosphonophenylglycine). However, the existence of a single receptor protein type should not rightfully allow for the observed distinction in the receptor kinetics. The original authors ruled out the significant contribution of voltage-gated channel activity in the observed transient/sustained distinction due to the lack of changes experienced under voltage clamp recordings. The variety in ON BC response kinetics, therefore, could rather be attributed to variations in the downstream signaling routes between the mGluR_6_ receptor and the final target, the TRPM1 receptor [10,53,54,55,56], and/or to the distinct inhibitory effects present in the inner retinal network [57]. Therefore, albeit kinetics of postsynaptic receptors in the PR-to-BC synapse may play a role in shaping response transience for OFF BCs, it is undoubtedly not a significant factor for determining response decay in ON BCs. Consequently, by the time BC signals reach RGCs, response kinetics are transformed several times via various circuit elements and mechanisms [58]; thus, the output of the retina depends even less on the synaptic signal transfer through the postsynaptic glutamate receptors of BC dendrites. 

#### 3.1.2. Outer Retinal Inhibition

HCs are GABA-expressing inhibitory cells in the outer retina. They receive glutamatergic excitation from photoreceptors and provide feedback inhibition to photoreceptors and feedforward inhibition to BCs (Figure 4) [13]. It has been shown that the inhibitory feedback is established through ephaptic mechanisms whose morphological substrate is provided by pore-forming hemichannels [59,60,61]. According to an alternative hypothesis, the feedback is mediated by proton pumps that, upon illumination, change the pH in the synaptic cleft [62]. Either way HCs perform feedback signaling, it appears that this contact plays a major role in contrast enhancement and the generation of center-surround receptive fields [14,63,64,65,66]. The HC feedback to photoreceptors, however, is rather uniform throughout the entire retinal lattice, and therefore does not offer any explanation for the observed kinetic differences in the light responses of neurons in the downstream circuitry, including RGCs. Besides the feedback to photoreceptors, HCs also express GABA as a neurotransmitter that serves feedforward inhibition to BC dendrites [67,68,69,70,71] and, supplementing the local feedback to photoreceptors, underlies a widespread inhibitory mechanism for the circuit [67]. In addition, GABAergic transmission may also serve as a source for auto feedback used to set HC sensitivity through autoreceptors [61]. Although the feedback and feedforward contacts might contribute to type-specific differences in signaling through parallel retinal routes (e.g., quantitative differences and/or differential subunit composition of postsynaptic GABA receptors on BC dendrites), recent data show that RGC response transience was unchanged when HCs were disabled [72], thus indicating that HC signaling does not shape RGC response kinetics via any of the above-mentioned mechanisms (feedback to photoreceptors, feedforward to BCs, or auto feedback).

### 3.2. Bipolar Cell Characteristics

Beyond subtype-specific differences in the synaptic interactions at the first synaptic layer, the differential passive and active membrane properties of various BCs may also account for some of the observed kinetic variations in BC responses. Given that BCs are, apart from input specificity in the OPL and selective stratification in the inner plexiform layer (IPL), relatively similar in shape, one does not normally expect considerable differences in their passive membrane features. Since the biological membrane acts as a low-pass filter, dendritic excitatory postsynaptic currents (EPSCs) are expected to become more sustained as they travel from the dendrites to the axonal ending, regardless of the initial kinetics of the response. In fact, it has been shown that axotomized BCs display relatively sustained responses compared to those recorded from intact counterparts of the same BC subtype [49]. This suggests that EPSCs go through low-pass filtering [10] and they all reach the inner retina as relatively sustained signals, meaning that transient BC outputs are recreated in the axons of certain BC subtypes. Nonetheless, it is expected that differential active membrane properties and/or synaptic interactions (e.g., inhibitory feedback) of the various BC subtypes play a significant role in shaping BC response kinetics. 

#### 3.2.1. Active Membrane Properties of BCs

One possible source of the differential response kinetics and glutamate release can be accounted for by the expression of various voltage-gated Ca^++^ channels (CaV) that trigger transmitter release from vesicles [73] (Figure 5). However, the expression of these channels shows species-dependent differences rather than BC subtype specificity. Transient T-type (low voltage-activated) channels are expressed by all BC types in mice [74,75], while exclusively sustained L-type (high-voltage-activated) Ca^++^ channels are expressed in goldfish [75,76], and they are both present in various salamander and zebrafish species [77,78]. Since both transient and sustained BC subtypes exist in all species, it is likely that the CaV composition does not partake in shaping BC response kinetics (at least not transience). Moreover, it has been shown that both sustained and transient BCs display Ca^++^-spikes in the zebrafish retina [79], which also argues against the hypothesis that any difference in CaV expression could play a major role in determining response kinetics.

Interestingly, BCs that largely operate by maintaining and transmitting slow potentials appear to express voltage-gated Na^+^ channels (NaV) in various vertebrate species, including both cold-blooded animals [80,81,82] and mammals [47,83,84,85]. The presence of the fast NaVs allows for certain BC subtypes to generate all-or-nothing spikes to amplify synaptic events and to enhance transient BC response components. The expression of Na^+^ channels is prominent in transient BC subtypes [47], which further supports this latter view (Figure 5).

Finally, BC axon terminals have also been shown to express additional channels that may also contribute to shaping BC responses. For example, some BCs possess slowly activating, delayed rectifying voltage-gated K^+^ channels, whereas others express channels that rapidly activate and slowly inactivate instead [77,86,87,88,89] (Figure 5). Evidently, depending on the activation speed of the outward K^+^ current, the corresponding response component will be attenuated. In this scheme, slowly activated K^+^ currents make BC responses more transient, whereas rapidly activating currents by attenuating all response components results in more sustained BC responses. In addition, voltage-gated hyperpolarizing HCN channels may also exist in BC axon terminals [77,90,91] (Figure 5). These, when expressed, mediate faster recovery of responses from the prior activation. Finally, the existence of Ca^++^-gated Cl^−^ channels has also been demonstrated in BC axons [92] (Figure 5). The activation of this channel induces a shunt by increasing a Cl^−^ conductance upon the depolarization-driven Ca^++^ entry, thereby making BC responses shorter and more transient.

#### 3.2.2. Autoreceptors and Glutamate Transporters of BCs

There are still a few other factors that can influence the response kinetics of BCs. One of these is the expression of autoreceptors in the form of metabotropic glutamate receptors (more precisely, mGluR7) on the axonal membrane [58,93] (Figure 6). Upon depolarization, BCs release glutamate into the synaptic cleft, and some of the transmitter molecules, instead of binding to ionotropic glutamate receptors on the postsynaptic membranes, bind to mGluR7 located on the membrane of presynaptic BCs. Through an intracellular cascade, these receptors then initiate feedback to regulate glutamate release in both ON and OFF cone BCs [93]. Interestingly, this autoreceptor-mediated mechanism has been shown to convey positive feedback, making responses of BCs as well as responses of their postsynaptic partner RGCs and ACs more elongated and sustained [58,94]. It must be noted that this feedback mechanism seems to be subtype-specific, as ipRGCs and Hoxd10+ ON RGCs were under this control mechanism, whereas other examined cells (TRHR+ and Hox10+ OFF) were not [58].

In contrast to autoreceptors, glutamate transporters that are expressed in bipolar cell axonal terminals [95,96] perform a negative feedback mechanism to control glutamate release from BC axon terminals (Figure 6). However, ipRGC responses became smaller following a pharmacological blockade of glutamate transporters [58], contradicting the negative feedback hypothesis. This discrepancy in the literature can be explained with BC subtype-specific transporter expression and/or expression in synapses that primarily drive inhibitory ACs, resulting in the gross effect of the functioning glutamate transporters appearing to enhance some RGC responses [58].

### 3.3. Inner Retinal Contribution

Responses of axotomized BCs often differ from those of intact cells in slice preparation [49], indicating a heavy influence of inner retinal mechanisms on BC response kinetics. This finding has been supported more recently by demonstrating that outputs of a single BC can produce wildly different and type-specific RGC response kinetics, and that this diversity is mediated by ACs [97,98]. In fact, several inhibitory mechanisms in the inner retina have been suggested to shape BC responses prior to the information passing to RGCs [27,78,99,100,101,102,103,104,105,106], as discussed in subsequent sections. 

#### 3.3.1. AC to BC Feedback Inhibition

Much of the inner retinal interactions involve inhibition provided by glycinergic and/or GABAergic ACs (Figure 7a). Some of these inhibitory mechanisms are local and operate by the classic schematic of inhibitory feedback. The best-known example of this phenomenon is established by the S1/S2 ACs, A17 cells in cats [107,108,109,110,111,112,113,114] that receive excitatory signals from rod BCs and provide local GABAergic inhibition to the same rod BC terminal [112,115,116]. It has been established that this particular feedback mechanism acts via a combination of GABA_A_ and GABA_C_ receptors [117,118,119,120,121,122,123] and has a profound effect on the temporal properties of rod BC responses as well. The relative sluggish Cl^−^ currents of GABA_C_ receptors appear to accelerate the response decay, thus making rod BC responses more transient [117]. Similarly, relative late inhibitory signal components make light-evoked responses of some BCs more transient in the cone system as well [124]. On the other hand, GABA_A_-mediated negative feedback to the BC terminals induced more sustained responses of ON alpha RGCs in the mouse retina [125].

#### 3.3.2. AC to RGC Feedforward Inhibition

Apart from feedback contacts onto BC terminals, other forms of inhibitory mechanisms also exist in the inner retina, including feedforward inhibition to inner retinal neurons (to spatially offset BCs, ACs, and to RGCs; Figure 7b). The inhibitory effect on response transience depends on the delay of the inhibition-driven outward current relative to the BC-mediated excitatory input on postsynaptic cells. Each RGC subtype receives a different relative contribution of three forms of feedforward inhibition, including local glycinergic, local sustained GABAergic, and broad transient GABAergic inhibition [126]. It appears that broad transient GABAergic inhibition has the shortest latency, local glycinergic inhibition has an intermediate latency, whereas local sustained GABAergic inhibition possesses the longest latency. According to this general scheme, sustained GABAergic inhibition mainly acts on late response components, resulting in more transient RGC responses. On the contrary, quicker transient GABA- and glycine-mediated currents attenuate the early phase or the entire course of the response, respectively. Expectedly, these latter two inhibitory mechanisms truncate RGC responses and thus make them sustained. Some of the circuits that partake in the aforementioned inhibitory mechanisms have already been discovered, diversifying BC inputs by either prolonging or shortening RGC responses. One such example is the population of the vasoactive intestinal polypeptide (VIP)-expressing AC that, by providing quick GABA-mediated inhibition, makes ON polarity RGC responses more prolonged by truncating the early peak excitatory component of their response [33]. In addition, knowing that these cells belong to the so-called wide-field population, the provided inhibition is very likely not local, but far-reaching instead [127,128,129,130,131]. In contrast, relatively delayed feedforward inhibition shortened and attenuated responses of ON midget cells in the primate retina [132], resulting in inhibitory input that is delayed relative to BC-driven excitation to midget RGCs.

#### 3.3.3. Crossover Inhibition

A third common inhibitory motif that takes place in the inner retina is the so-called push–pull or ’crossover’ inhibition (Figure 7c). In this case, increases in excitatory input translate into an inhibitory drive onto opposite polarity cells in the same region. This involves ACs with diffuse or bistratified dendritic arbors that allow for the collection of excitatory signals with one set of dendrites and provide inhibitory output through the other group of dendritic branches. Crossover inhibition has been shown to modify the kinetics for most OFF and about half of the ON polarity bipolar cells in the vertebrate retina [124]. The best-known AC that provides crossover inhibition is the narrow-field AII AC. AII cells receive excitation from the ON polarity rod BCs and, in turn, provide glycinergic inhibition [133,134] to OFF polarity cone BCs in the same local retinal area [135,136,137,138,139]. Thus, whenever signals are conveyed by the ON polarity rod BC, glycinergic inhibition will be generated on nearby OFF cone BC axon terminals. As a result, the simultaneous activation and attenuation of ON and OFF signaling pathways occurs, respectively. The existence of other similar crossover inhibitory mechanisms by which ACs either carry ON inhibition to OFF cells or OFF inhibition to the ON cells, although not as well described as the one provided by AII cells, has been suggested [140]. It has been shown that crossover inhibition compensates for the predominantly non-linear BC to RGC synaptic mechanisms by mediating sustained neuronal responses prerequisite for linear signaling [140,141]. Linear signal processing, in turn, is important for certain RGCs as it is required for computing various properties of the visual world, such as average intensity across a receptive field and maintaining the distinction between brightness and contrast [140]. Crossover inhibition has been shown to have surprisingly little impact on the responses of primate ON parasol RGCs when simple stimulation was utilized [132]. However, spatially structured stimulation caused a simultaneous elevation of excitatory and inhibitory inputs to ON parasol cells, after which inhibitory inputs substantially abbreviated the RGC spike output.

#### 3.3.4. AC to AC Inhibition

Finally, ACs that provide all of the above-mentioned inner retinal inhibitory mechanisms are, in fact, under inhibitory control themselves (Figure 7d). This, in turn, modulates the strength of the inhibition (feedback, feedforward, and crossover) they provide onto other inner retinal neurons. These disinhibitory, serial inhibitory, and nested inhibitory mechanisms [142] fine-tune the spatiotemporal characteristics of direct inhibitory action on BC and RGC processes and partake in determining the response kinetics of all inner retinal neurons, including RGCs. The most common interaction between ACs appears to be the crossover inhibition, and the majority of ON ACs receive glycinergic OFF inhibition and about half of the OFF ACs receive glycinergic ON inhibition [143]. Interestingly, GABAergic inhibition seems to play only a minor role in this process.

### 3.4. Summation of Signals in the Inner Retina

If BC subtypes that feed RGCs were dominant in determining and shaping RGC response kinetics, then one would expect that all RGCs maintain comparable response transience when the same single BC type dominates their inputs. In contrast to photopic signals delivered by 5–6 bipolar cells to ON RGCs, low scotopic signals reach RGC targets mostly via the primary rod pathway that operates with the single rod BC type. Therefore, the same inputs are delivered to all RGC subtypes under scotopic conditions. In contrast, our yet unpublished data show that the scotopic RGC responses are just as diverse in terms of their kinetics as their photopic counterparts (Figure 8). This suggests that the observed kinetic variety of RGC responses is neither the result of the differential kinetics of the phototransduction cascade of rods and cones or the postsynaptic glutamate receptor on BC dendrites. In addition, different experiments demonstrated that outputs of a single BC can produce widely different RGC response patterns (including differences in kinetics), depending on the RGC type, and that this diversity is mediated by retinal ACs [97,98]. This suggests that irrespective of the kinetic nature of BC inputs, RGC response kinetics are likely reestablished in the inner retina both by passive and active membrane properties of RGCs. While circuit elements (mostly inhibitory) have been described in the previous section, there are a few more aspects to cover in this regard. These include the potential role of desensitization of postsynaptic glutamate receptors on RGC dendrites that may play a role in the generation of transient RGC responses [144], and also other voltage-gated currents that shape RGC responses [145].

Apart from RGC membrane properties, there is another crucial but often neglected factor that determines RGC response transience, namely the summation of all incoming inputs to the same RGC. One aspect of this issue, the feedforward inhibition from ACs to RGC, has already been discussed (see above in Section 3.3.2). However, RGCs may also integrate excitatory signals from more than one BC. Figure 8 provides one example (our unpublished data), where an examined RGC clearly displays relative transient responses to both scotopic and photopic stimuli (small PSTHτ values, for the calculation of PSTHτ, refer to [146]), but the full-field light response of the same cell appeared more sustained (larger PSTHτ values) in the mesopic/low photopic range when both scotopic and photopic signaling streams were active. This is obviously the result of the summation of signals carried via a separate cohort of BCs serving at least two different signaling pathways. Under scotopic and photopic conditions, both the PSTH peak and τ values were influenced by only one signal component (the slow and sensitive component in scotopic conditions and the fast but less sensitive component in photopic conditions), whereas mid-range illumination brought about an intermingling of the fast signal component peak and a shoulder of the slow response component, thereby resulting in an increased PSTHτ value. The summation of these signals evidently caused the spiking pattern of this otherwise transient cell to appear more sustained. A similar signal summation should occur as well when two or more photopic signals are conveyed parallel to each other to the same RGC targets. In this scheme, a certain RGC subtype receives inputs from two or more signaling streams (BCs). This, in fact, is the case for ON alpha RGCs of the mouse retina that receive a mixture of inputs from type 6, 7, and 8 bipolar cells [147]. In addition, sustained OFF alpha cells are postsynaptic to type 2 bipolar and GluMI cells, whereas transient OFF alpha cells are targeted by type 3a and type 4 bipolar cells [148,149]. Therefore, the mixing of inputs originating from several parallel signaling streams seems to be a general feature for most RGCs in the mammalian retina. Furthermore, it appears that the summation of inputs with slightly different kinetics (delay and decay) and sensitivity plays a crucial role in determining the ultimate kinetics of RGC responses. Again, these results are not necessarily in conflict with other reports regarding mechanisms that determine BC response kinetics, however BC response characteristics are not simply inherited by postsynaptic RGCs but greatly altered via signal summation. The results of such signal summation can be exemplified for some RGC subtypes that have been studied more deeply, including the sustained and the transient OFF alpha cell populations. Sustained OFF alpha RGC responses are generated by the summation of excitatory inputs from the transient type 2 bipolar cells and GluMI cells [43,148,150], whereas the synaptic partners transient OFF alpha cells are the transient type 3a and sustained type 4 bipolar cells [149,150]. 

Clearly, rather than simply inheriting response kinetics of presynaptic BCs, these two RGC subtypes performed a transformation of incoming signals. One issue remains: while the summation of excitatory signals can provide an example of how transient inputs are transformed into intermediary or sustained signals by RGCs, the reverse (sustained-to-transient transformation) cannot be explained by the same mechanism. It is known that in addition to excitatory BC inputs to RGCs, some signaling routes also provide inhibitory signals via intermediary ACs. In fact, the sections above detailed how GABAergic inhibition may alter RGC response transience. In this scheme, the early onset of inhibition truncates the response of the RGC target, thus making it more sustained. GABAergic AC input alters RGC response kinetics and functionality, as a fast inhibition is known to be capable of truncating excitatory signals [33,151], whereas delayed inhibitory inputs can shift the signal towards the more transient domain of the spectrum (Figure 9) [49,121,152].

## 4. The Visual Function of RGC Transience and Future Directions

The transient/sustained division of RGC light responses has been widely utilized to characterize and classify RGC subtypes, and it is thought to be strongly related to visual function. Indeed, it has been reported recently that response delay, another important kinetic feature of RGC responses, is subtype-specific, and it is precisely fine-tuned by inner retinal microcircuits to achieve better performance for vision [25]. Therefore, it is not a far-fetched idea to expect that RGC response transience also contributes to visual coding. Accordingly, it is generally accepted that transient, burst-like responses likely transmit information about ‘fast-paced’ and dynamic aspects of the visual field, including direction and movement, whereas sustained responses provide a continuous feed of information on the static elements. Therefore, transient and sustained RGC responses encode dissimilar but equally important facets of visual information. This functional divergence already starts at the level of BCs, as sustained type 1 BCs mediate color vision for the OFF polarity signaling stream, whereas other, more transient cells do not [153]. The increasing number of 40+ different types of mammalian RGCs [6,153,154] cover a rather wide range of response transience, suggesting that they can be specialized to perform a large variety of visual tasks. There has been a large collection of evidence supporting this view, including RGCs with transient responses that encode object movement [16,155,156,157], the direction of motion [158,159,160,161], and also others with sustained responses that perceive luminosity contrast [162], color contrast [163], or object orientation [164]. While the first cohort of RGCs require a quick inactivation and a corresponding decay of spiking frequency in order to quickly recover and get ready for the following changes in the visual scene, sustained RGCs allow for the summation of inputs over an extended timeframe in order to become more sensitized for minuscule differences of light levels (e.g., grayscale or color) within their receptive fields. Additionally, it seems that RGC subtypes with larger dendritic and receptive fields exhibit greater spatial but briefer temporal integration and higher gain, while on the other hand, more sustained RGCs have smaller arbors in general [165]. Moreover, transience is not a static feature of retinal neurons, but it can be dynamically changed by utilizing dedicated inner retinal circuits to actively adjust the response length (and also other kinetic features) to match the desired physiological function. Furthermore, it has been shown that axotomized rod BCs displayed a considerably reduced dynamic range relative to their healthy partners [49], thus indicating that in inner retinal circuits, preferably inhibition is utilized to adjust response transience and dynamic range. It is possible that the dynamic range of transient RGC responses, similar to BCs, surpasses that of sustained ones. In fact, Ravi and colleagues [165] found that RGCs with transient responses (brief temporal integration) display greater dynamic ranges.

One interesting aspect of this hypothesis arises when adaptation- and/or stimulus-dependent changes in RGC response transience occur. Does that mean that RGCs perform better in certain stimulating conditions than in others? The answer to this question is, not too surprisingly, yes. We experience this effect on an everyday basis: our vision is rather limited during the night, and this limitation involves the reduction of contrast sensitivity in both the spatial and temporal domains of our vision. This latter phenomenon is expressed by the Ferry–Porter Law, stating that the critical fusion frequency is proportional to the logarithm of the flickering stimulus luminance [166]. Therefore, the precise adjustment of RGC response temporal features, including transience, appears to be critical for our visual perception, and the mammalian retina has specific circuits, mainly in the inner retina, that are responsible for fine-tuning this kinetic feature to match the output signal of RGCs to match their subtype-specific function. Besides visual neuroscience, these conclusions might provide useful information for algorithms of computer vision or next-generation retinal prostheses as well.

## Figures and Tables

**Figure 1 cells-11-00810-f001:**
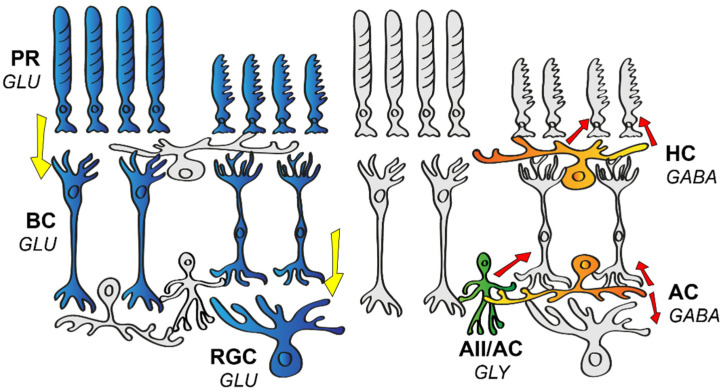
Schematic drawing of the basic neuronal wiring of the mammalian retina. The vertical information stream encompasses the light-sensitive photoreceptors (PR), bipolar cells (BC), and retinal ganglion cells (RGC). These cells express glutamate (GLU), thus providing an excitatory (highlighted in blue) stream of signals (direction of information flow is indicated by yellow arrows). Horizontal cells (HC) and amacrine cells (AC) serve various forms of inhibition (red arrows) in the outer or the inner retina, respectively. HCs express GABA (highlighted in orange in the outer retina), whereas ACs release either glycine (GLY, highlighted in green) or GABA (highlighted in orange in the inner retina) as a neurotransmitter.

**Figure 2 cells-11-00810-f002:**
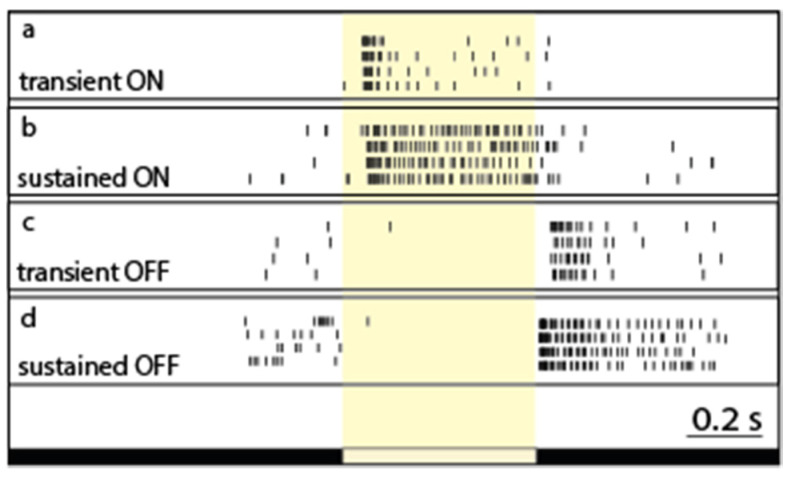
The diversity of response transience across the retinal ganglion cell population. Peri-event rasters of representative RGCs (**a**–**d**). Light-evoked spiking responses upon full-field illumination are rather similar across trials for each RGC, but they display a great variety in terms of their response length (or decay—expressed as the PSTHτ value in this work) for both the ON (cells 1 and 2) and OFF (cells 3 and 4) subpopulations. The white bar below the recordings represents the timing of the on- and off-set of the stimulus.

**Figure 3 cells-11-00810-f003:**
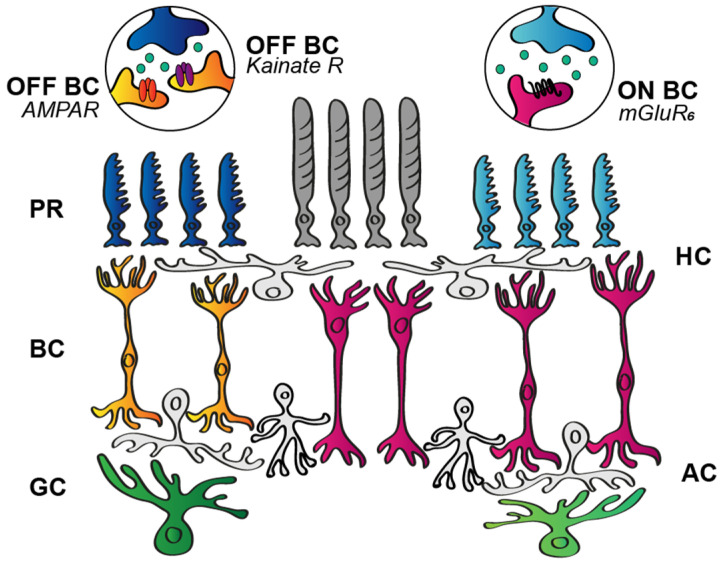
Excitatory signal processing in the outer retina. Photoreceptors (PR; dark grey, dark and light blue) release glutamate that binds to postsynaptic glutamate receptors located on dendrites of bipolar cells (BCs). OFF BCs (orange) express either AMPA or kainate ionotropic receptors that, upon glutamate binding, excite BCs. In contrast, ON BCs (magenta) have the metabotropic mGluR6 receptor on their dendrites and, unlike ionotropic receptors, hyperpolarize ON BCs when glutamate binding occurs (note that rods also form the same type of contact with the rod BCs). Horizontal cells (HC), amacrine cells (AC), and ganglion cells (GC, green) are also represented in the retinal circuitry.

**Figure 4 cells-11-00810-f004:**
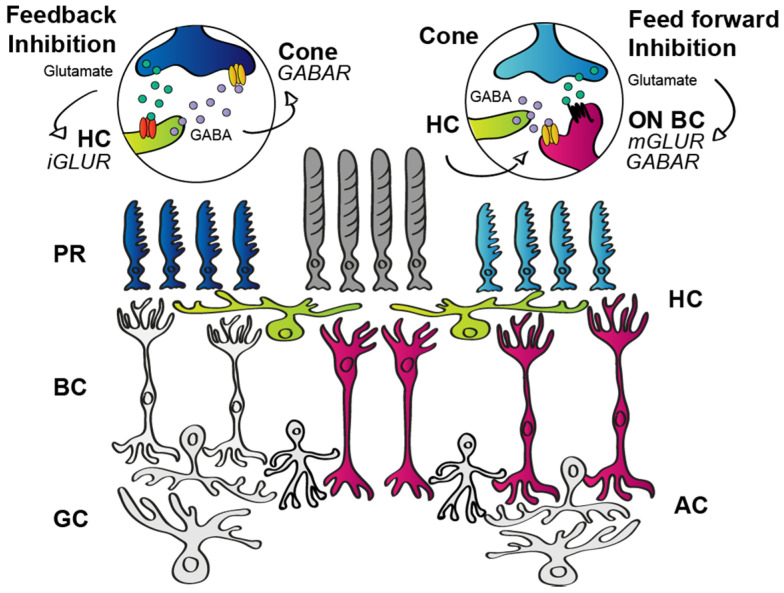
Inhibitory mechanisms in the outer retina. Photoreceptors (PR; dark grey, dark and light blue) release glutamate that binds to ionotropic glutamate receptors located on the postsynaptic dendritic surfaces of horizontal cells (HC, green). HCs in turn release GABA as an inhibitory transmitter and provide feedback inhibition to PRs (left side of the scheme—note that HC to rod BC feedback inhibition is not shown in the diagram but exists as well) and feedforward inhibition to BCs (magenta—right half of the scheme). Amacrine cells (AC) and ganglion cells (GC) are also represented in the retinal circuitry.

**Figure 5 cells-11-00810-f005:**
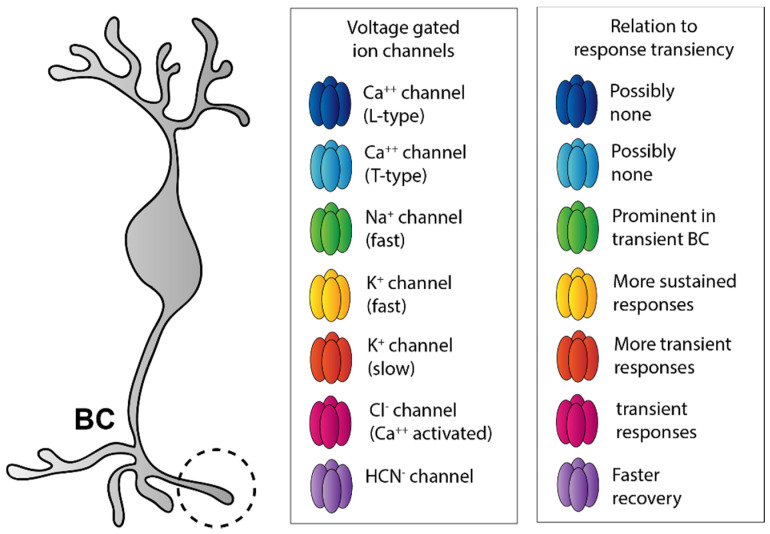
Active membrane properties of bipolar cells. Most voltage-gated channels that determine active membrane properties of BCs and potentially shape the transience of light responses can be found in the axon terminal region (left panel). These include various Ca^++^, K^+^, Na^+^, and Cl^−^ channels (middle panel), some of which increase, whereas others decrease the BC response transience (right panel). Color codes in the middle and right panels connect each channel type with a function they play in regulating BC response transience.

**Figure 6 cells-11-00810-f006:**
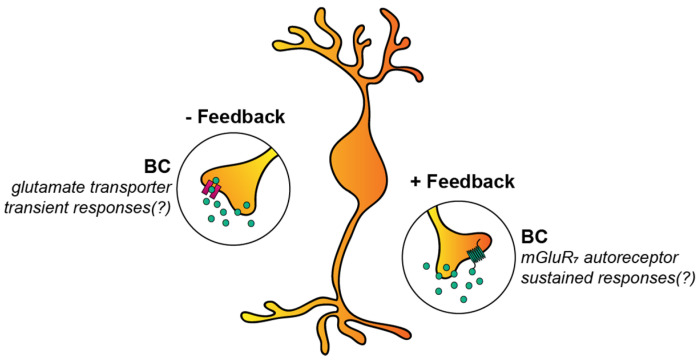
Self-regulation of bipolar cell responses. There are two identified mechanisms by which retinal bipolar cells (BCs) likely self-regulate their light responses. One of them takes place via glutamate transporters that sequester glutamate from the synaptic cleft, and thereby the effects of the transmitter and the light response become more transient (left panel). A second mechanism occurs through mGluR7 receptors that bind some of the released molecules and exert their effect on the dynamics of release and the light response itself (right panel). The net effect of the activation of mGluR7 receptors in the BC axon terminals makes BC responses more sustained (also see the text).

**Figure 7 cells-11-00810-f007:**
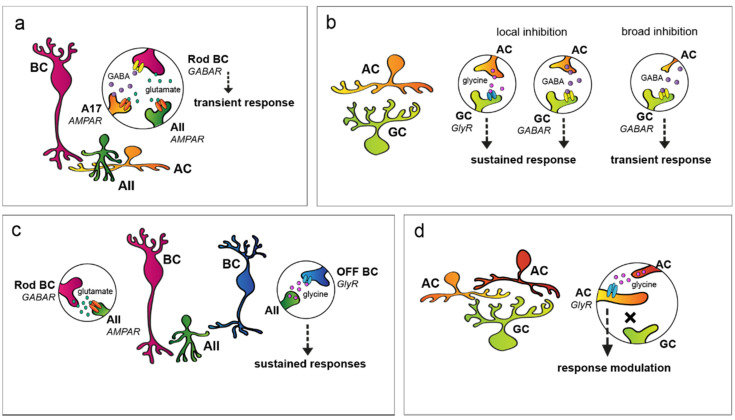
Inhibitory mechanisms in the inner retina. (**a**) Upon glutamate release from rod bipolar cells (BCs, magenta), A17 amacrine cells (ACs, orange) provide a relative sluggish GABAergic inhibitory feedback to rod BCs and make BC and postsynaptic AII AC (green) responses more transient. (**b**) ACs also provide feedforward inhibition to retinal ganglion cells (RGCs) in the inner retina. Both glycinergic and GABAergic interactions result in more sustained RGC responses in certain circuits (two middle panels), whereas the activation of some GABAergic contacts results in more transient RGC responses (rightmost panel). (**c**) Certain ACs in the retina, upon activation, provide opposite polarity inhibition to inner retinal neurons. In the presented circuit, AII ACs (green) receive glutamatergic excitation from rod BCs (magenta) that have ON polarity, and in turn, the AII cell forms inhibitory glycinergic synapses with opposite polarity OFF cone BCs (blue). This type of interaction is called crossover inhibition. (**d**) ACs (red) often make inhibitory synapses with other ACs (orange). Via such disinhibitory mechanisms, the transience of the final output RGCs (green) can be finely tuned to suit certain visual functions.

**Figure 8 cells-11-00810-f008:**
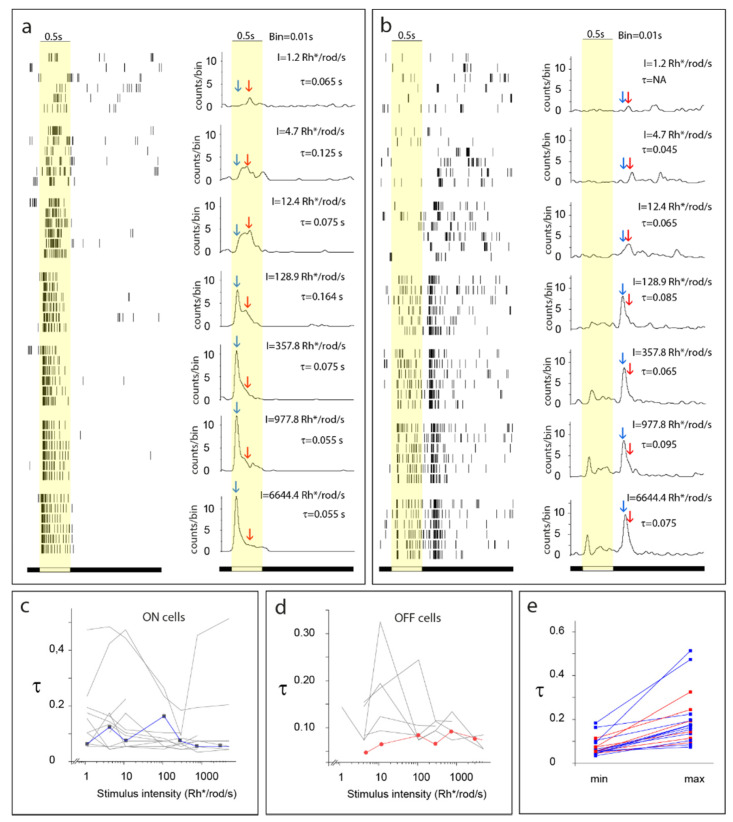
Signal interference through parallel retinal pathways. (**a**) Representative ON retinal ganglion cell (RGC) light-evoked rasters (left column) and peristimulus time histogram (PSTH) cohorts (right column) were obtained to light stimuli of various strength (intensity is reflected in the right top corner). RGC PSTHτ values (for more information, see [146]) changed non-monotonously during this experiment while the stimulus intensity was gradually increased (see also panel (**c**)). PSTHs clearly show a very sensitive, relatively delayed response component (red arrow) and a less sensitive but fast response component (light blue arrow). These two components differ in their delays but appear similar in response decay, and therefore PSTHτ values are shifted towards the sustained range when the two signals are summated (mesopic conditions—2nd, 3rd, and 4th panels), whereas they appear transient when only one signal is present (scotopic condition—1st panel) or dominates over the other component (photopic conditions—5th, 6th, and 7th panels; see also panel (**c**)). (**b**) A similar experiment was performed for this representative OFF RGC. PSTHτ values of this RGC clearly changed during this experiment when the stimulus intensity was gradually increased. This OFF RGC also showed a very sensitive and delayed response (red arrow) as well as a faster but less sensitive (light blue arrow) response component. The two signal components differed in their delays and sensitivities and a slight alteration in PSTHτ values occurred as a result of the summation of components (mostly in mesopic conditions—middle panels). While the distinction of response components can clearly be differentiated for the ON RGC in (a), this OFF cell (and most examined RGCs) showed a less obvious and less separable summation of incoming signals. (**c**,**d**) Diagrams show that similar to cells shown in panels (**a**,**b**) (values of these cells appear in blue and red in the diagrams), most recorded RGCs displayed stimulus strength-driven changes of PSTHτ values (grey curves). (**e**) The diagram shows minimum/maximum PSTHτ value pairs for the recorded ON (blue) and OFF (red) cells during the course of the stimulus intensity recording paradigm. The examined RGCs showed ~18–73% PSTHτ changes during the course of this experiment.

**Figure 9 cells-11-00810-f009:**
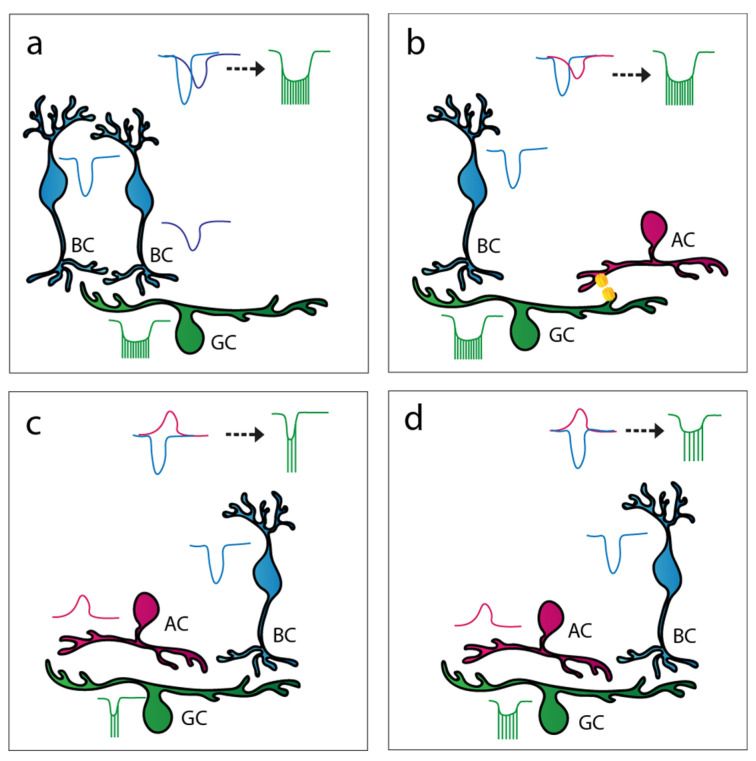
Summary of potential signal summation mechanisms affecting RGC response transience. (**a**) Two bipolar cells (BCs) of different subtypes provide transient inputs to the same retinal ganglion cell (RGC; light blue EPSC curves). These two inputs have dissimilar delays (due to differential BC signaling and/or a different location of synapses over the RGC dendritic arbor), and therefore the summation of the responses results in an intermediate or sustained RGC spiking response. (**b**) This RGC receives excitatory inputs from two sources: from a transient BC (light blue EPSC) and from a gap junction-coupled amacrine cell (AC; purple depolarization). If the dynamics of these two inputs differ, their summation will induce intermediate and/or sustained RGC spiking. (**c**) This RGC receives excitation from a BC (light blue EPSC) and delayed inhibition (red IPSC) from an AC, resulting in a transient RGC response. (**d**) An RGC that receives excitation from a BC (light blue EPSC) and inhibition (red IPSC) from an AC. In this scenario, the two inputs have about the same delays, and therefore the excitation will be truncated and the RGC output is an intermediate/sustained spiking.

## Data Availability

Not applicable.

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
