# Peer review of "Transience of the Retinal Output Is Determined by a Great Variety of Circuit Elements"

_cells, 2022, doi:10.3390/cells11050810_

Round 1
Reviewer 1 Report
The authors provide a timely review about the origins of the response kinetics – transient/sustained - in retinal ganglion cells. First, let me congratulate the authors to review such a complex topic. They provide a detailed summation of the cellular origins of temporal kinetics in the retina from different species. They also showcase unpublished data to underline the importance of this topic. Though the review is well worked out and I endorse publication, it needs some minor correction.
Minors:
- Keyword start with gap junctions, connexins… But they are barely reviewed here. Consider better key words.
- Line 23: >30 RGC types…in Introduction states 46 RGC types. Why not state 46 types?
- Line 50: 3 distinct photoreceptors – rod, cone and what else? You mean rod, S- and M cones ? This should be a bit clearer formulated.
- Figure 1 can improved. The HC is lacking their axon terminal that connects to rods – unique feature of mouse HCs. Suggestion: Use A and B as panel indicator to clearly separate between glu and gaba or maybe make just one overview? Another suggestion: add arrows to make clear in which direction the glu and gaba transmission goes. It would probably also help to indicate the ON and OFF pathways which are later mentioned.
- Line 79:80: what does (BCs:1(ab),2,3(a,b),4) and (5(a-80 d),6,7,8/9) mean?
- Line 83: HC Feedback update Curr Biol. 2022 Feb 7;32(3):545-558.e5. doi: 10.1016/j.cub.2021.11.055. The article should be included
- Line 89: Eggers paper is from 2011 - is there something newer?
- Line 91: why bringing up epigenetic when it is not further discussed?
- Line 126-134 ff: It is a strong – and in my opinion not correct - statement that there is no model available. There are publications investigating the origin of transience in retinal ganglion cells e.g. Euler et al 2014 and other mentioned later. Consider toning it down.
- Line 148: 10-14 cone bipolar cells vs 15 in chapter 1 (line 71). The numbers should be consistent through the manuscript
- Line 160 ff: The Awatrami and Slaughter paper [40] is not further mentioned here and comes later in line 200ff. Consider discussing it when mentioned.
- Figure 3 legend: green colour for RGC not mentioned. All PRs release glu in the dark – not only cones as indicated by blue colours. You could change photoreceptors to cones (for all legends).
- Line 220 ff Same as above: HC Feedback update Curr Biol. 2022 Feb 7;32(3):545-558.e5. doi: 10.1016/j.cub.2021.11.055. The article should be included.
- Figure 5 and 6 are identical – I think that should be changed.
- Figure 5 (and 6 ) legend right box: change “rransient” and “ransient” to transient.
- Line 268: The paragraph starts with that voltage gated calcium channels are responsible for response kinetics but in the Figure 5 box they are mentioned not to play a role. Or did I get that wrong?
- Chapter 3.3 and its subchapters: I am getting a bit lost here. You may want to add a line at the end of each subchapter how that all relates to transience.
- Figure 8 legend: It is too long. No need to describe in detail all the rasters – summarize them.
- Line 487: “obviouslythe” to obviously the
- Line 567: Figure 8 should be Figure 9
Up to the authors and I don’t expect a response: Consider leaving chapter 1 out. All other chapters describe in detail the wiring of the retina i.e. the specific discussed part. No need to introduce a general wiring. It also would solve my first minor points raised here
Author Response
We thank the reviewer for the suggestions that made our manuscript even more thorough and also for pointing out mistakes that now are corrected in the revision.
- Keyword start with gap junctions, connexins… But they are barely reviewed here. Consider better key words.
The reviewer is absolutely right, we replaced these two terms with new ones that are more appropriate.
- Line 23: >30 RGC types…in Introduction states 46 RGC types. Why not state 46 types?
We felt that giving an exact number requires the citation in the abstract and we wanted to avoid that. For that reason we chose to give an approximation. However, upon the referee request we changed this to ~46.
- Line 50: 3 distinct photoreceptors – rod, cone and what else? You mean rod, S- and M cones ? This should be a bit clearer formulated.
We chose simlicity and changed the number to 2 instead of explaining the two spectrally different cone types (which would stand only for the dichromat species anyways).
- Figure 1 can improved. The HC is lacking their axon terminal that connects to rods – unique feature of mouse HCs. Suggestion: Use A and B as panel indicator to clearly separate between glu and gaba or maybe make just one overview? Another suggestion: add arrows to make clear in which direction the glu and gaba transmission goes. It would probably also help to indicate the ON and OFF pathways which are later mentioned.
Than you for the suggestion! Out of the two options we chose the second one and placed arrows in the figure to indicate the information flow for bot excitation and inhibition. We also altered the corresponding figure caption.
- Line 79:80: what does (BCs:1(ab),2,3(a,b),4) and (5(a-80 d),6,7,8/9) mean?
These are the names of the various BC subtypes in the indicated references (Morgans et al. 2009; Puthussery et al. 2014).
- Line 83: HC Feedback update Curr Biol. 2022 Feb 7;32(3):545-558.e5. doi: 10.1016/j.cub.2021.11.055. The article should be included
Thank you for the suggestion! This paper is very relevant indeed, so we added it to both the text and the Reference list.
- Line 89: Eggers paper is from 2011 - is there something newer?
There are plenty of papers describing inner retinal GABAergic inhibition, many of those are newer than the cited Eggers’ paper. We cite those in later sections where we detail various forms of inhibitory circuits and their effects on RGC response transience.
- Line 91: why bringing up epigenetic when it is not further discussed?
We agree, having the word ’epigenetic’ was a bit out of the context, so we deleted it from the text.
- Line 126-134 ff: It is a strong – and in my opinion not correct - statement that there is no model available. There are publications investigating the origin of transience in retinal ganglion cells e.g. Euler et al 2014 and other mentioned later. Consider toning it down.
We toned down the statement and also cited the Euler paper here too.
- Line 148: 10-14 cone bipolar cells vs 15 in chapter 1 (line 71). The numbers should be consistent through the manuscript
Yes, there are 15 BCs altogether (rodBC and coneBCs together) in the mouse retina, while there are 10-14 cone bipolar cells described in the mouse and other mammalian species. However, most papers we cited were describing the mouse retina therefore we had to agree with the referee and changed 10-14 to 14 in the text.
- Line 160 ff: The Awatrami and Slaughter paper [40] is not further mentioned here and comes later in line 200ff. Consider discussing it when mentioned.
We do mention Awatramani and Slaughter in this sentence as we talk about rabbit and salamander retinas. In their paper, the authors utilized salamander as a model species. For that reason we left this sentenced unaltered.
- Figure 3 legend: green colour for RGC not mentioned. All PRs release glu in the dark – not only cones as indicated by blue colours. You could change photoreceptors to cones (for all legends).
We put a lots of thoughts to this question and considered the solution the referee suggested as well. Finally, we decided just to alter the figure caption by adding the color code (dark grey) to the cone color codes so now they are included. We also added a sentence to this figure legend explaining that the rod/rBC contact is the same as the cone/ON cBC contact. We performed similar changes in the legend of Figure 4.
- Line 220 ff Same as above: HC Feedback update Curr Biol. 2022 Feb 7;32(3):545-558.e5. doi: 10.1016/j.cub.2021.11.055. The article should be included.
We put the extra citation here too.
- Figure 5 and 6 are identical – I think that should be changed.
Yes. Many thanks for finding this error! We definitely made this mistake while inserting figures into the Cells’ standard manuscript sheet. In order to correct it we replaced Figure 6 to the schematic that intended to be there originally. Figure 5 was in the right place.
- Figure 5 (and 6 ) legend right box: change “rransient” and “ransient” to transient.
Thank you for noticing! We corrected the box in Figure 5.
- Line 268: The paragraph starts with that voltage gated calcium channels are responsible for response kinetics but in the Figure 5 box they are mentioned not to play a role. Or did I get that wrong?
The box to the right explains how each of the examined voltage gated channels change response transience. The referee is right that some of them does not seem to alter it (L-type Ca++-channels) while other ones are possibly play a role (T-type Ca++-channels) or known to alter it (make it more transient – Na+-channels, slow K+-channels, Cl—channels; make it more sustained – fast K+-channels). These changes are all indicated in the box and correspond to details in the text.
- Chapter 3.3 and its subchapters: I am getting a bit lost here. You may want to add a line at the end of each subchapter how that all relates to transience.
We point this out in chapter 3.3.1. lines 353-356 in the revision. For the chapter 3.3.2. lines 367-371 and also lines 374-376 describe changes of RGC response transience due to feed-forward inhibition. For the cross-over inhibition, the relevant lines are 398-400 and 405-408. For the topic of disinhibition the describing lines are 413-415.
- Figure 8 legend: It is too long. No need to describe in detail all the rasters – summarize them.
We agree, the figure legend for Figure 8 was rather long. We shortened it somewhat while trying to keep as much information content as possible.
- Line 487: “obviouslythe” to obviously the
Thank you for finding this typo! We fixed it.
- Line 567: Figure 8 should be Figure 9
Correct, thank you for finding this! We changed the legend to Figure 9.
- Up to the authors and I don’t expect a response: Consider leaving chapter 1 out. All other chapters describe in detail the wiring of the retina i.e. the specific discussed part. No need to introduce a general wiring. It also would solve my first minor points raised here.
We left chapter 1 in the manuscript because we think that it helps potential readers from other fields to follow descriptions of later chapters.
Reviewer 2 Report
In this review Ganczer et al detail the different retinal elements that contribute towards generation of transient output responses. The review is comprehensive as it details several cellular and circuit mechanisms in the outer and inner retina that could contribute towards shaping the temporal profile of visual responses (sustained vs transient). The figures and illustrations are well-suited to underscore the points being detailed in the text. Overall, this review will be an interesting read for visual neuroscientists. There are the following concerns with the current version of the manuscript which need to be addressed:
- The current version of Figure 6 is a duplicate of Figure 5. The Figure file thus needs to be amended such that the figure can match the current legend for Figure 6.
- Discussion of the AC to BC inhibition that shape BC output/response profiles are currently centered on GABAC receptors (section 3.3.1) alone and this should be expanded to include GABAA receptors as well. Specifically, considering discussion of A17 amacrine cells, Singer and Diamond 2003 and Chavez et al., 2006 have shown that GABA released from A17 amacrine cells at reciprocal synapses activates GABAA receptors. In fact, eliminating GABAA receptors from BC terminals has a direct impact on the kinetics of ganglion cell responses and temporal filtering of visual signals (Nagy et al., elife 2021).
- Discussion of the intrinsic delay between rod and cone photoreceptor responses (due to differences in the rod vs cone phototransduction cascade) is not considered - specifically in consideration of the discussion related to Figure 8.
- Discussion of the possible mechanisms by which HCs mediate feedback/inhibition in the outer retina (section 3.1.2) should also incorporate the possibility of proton (pH)-mediated feedback (Kramer and Davenport 2015).
- Figure 8 – it would be informative if the plot in panel e delineates which GCs are ON and which are OFF for the reader to get a sense of which types have a steeper min-max curve (i.e., greatest difference in temporal profile of response across luminosities).
- Page 6 line 238: Chaya et al., (cited reference #70) used a transgenic strategy (DTA line crossed to a HC specific Cre) to deplete HCs from the retina so the use of the phrase “when HCs were optokinetically disabled” is not accurate and needs to be amended.
- Page 5 line 203: the discussion of the “subunit composition” of mGluR6 is confusing as this metabotropic receptor is a seven transmembrane domain protein without canonical “subunits”. Clarification of the same would benefit the reader.
- The reference for Ganczer et al 2017 (page 12 line 254, page 13 line 485) is not included in the references section.
- Page 8 lines 309-311, the reference for mglur7 expression at BC terminals should not be Brandstatter et al 1997 (reference #91 detailing Kainate receptor distribution in the retina) but should instead be Brandstatter et al 1996 as this paper titled “Compartmental localization of a metabotropic glutamate receptor (mGluR7): two different active sites at a retinal synapse” shows data that suggests that mGluR7 is expressed at terminals of some cone bipolar cells.
- Page 3 line 101: the phrase “where an S-ON/M-OFF” needs to be completed.
- There are several typos in Figure 5: “ransient” or “rransient” instead of “transient” which need to be corrected.
- Legend of Figure 3 and 4: the photoreceptors have been denoted as “dark and light blue” but rod photoreceptors are depicted in grey.
- page 5 line 179 kainate is misspelled as kainite and in line 206 the alpha appears as a typo.
- page 7 line 266: “accounted for the expression” should read “accounted for by the expression”.
- page 11 line 445: The phrase “both by and passive/active properties” needs to be amended.
- page 13 lines 519-521: the sentence starting “ It is known…” needs to be amended to clarify its meaning.
Author Response
We thank the reviewer for the suggestions that made our manuscript even more thorough and also for pointing out mistakes that now are corrected in the revision.
- The current version of Figure 6 is a duplicate of Figure 5. The Figure file thus needs to be amended such that the figure can match the current legend for Figure 6.
Yes. Many thanks for finding this error! We made this mistake while inserting figures into the Cells’ standard manuscript sheet. In order to correct is we replaced Figure 6 to the schematic that intended to be there originally. Figure 5 was in the right place.
- Discussion of the AC to BC inhibition that shape BC output/response profiles are currently centered on GABAC receptors (section 3.3.1) alone and this should be expanded to include GABAA receptors as well. Specifically, considering discussion of A17 amacrine cells, Singer and Diamond 2003 and Chavez et al., 2006 have shown that GABA released from A17 amacrine cells at reciprocal synapses activates GABAA receptors. In fact, eliminating GABAA receptors from BC terminals has a direct impact on the kinetics of ganglion cell responses and temporal filtering of visual signals (Nagy et al., elife 2021).
Thank you for the additional reference! Indeed, these are highly relevant for the specific topic, thus we added them to both the text and the reference list of the manuscript.
- Discussion of the intrinsic delay between rod and cone photoreceptor responses (due to differences in the rod vs cone phototransduction cascade) is not considered - specifically in consideration of the discussion related to Figure 8.
We agree that the delay of rod and cone responses differ due to the mentioned differences in the phototransductional cascade (we can also see that in our recordings - like the RGCs shown in Figure 8). However, that is not necessarily result in the difference in the transience of rod and cone mediated responses, the subject of this review. Indeed, the response of selected RGCs showed that transience of rod- and cone mediated responses of the same cell might be very similar, they only alter transience when they occur together in mesopic conditions and due to the summation of slow and fast responses the kinetics are more sustained. For that reason, we did not put an extra chapter on rod and cone responses to this review, however, we added a few lines in the manuscript on this subject (lines 444-447).
- Discussion of the possible mechanisms by which HCs mediate feedback/inhibition in the outer retina (section 3.1.2) should also incorporate the possibility of proton (pH)-mediated feedback (Kramer and Davenport 2015).
We agree, the proton-pump hypothesis was missing from the manuscript, now we added an extra line to this section and cite the suggested reference in both the text and the reference list.
- Figure 8 – it would be informative if the plot in panel e delineates which GCs are ON and which are OFF for the reader to get a sense of which types have a steeper min-max curve (i.e., greatest difference in temporal profile of response across luminosities).
We made changes in the figure so ON and OFF RGCs are now color coded. We also performed alterations in the figure legend accordingly.
- Page 6 line 238: Chaya et al., (cited reference #70) used a transgenic strategy (DTA line crossed to a HC specific Cre) to deplete HCs from the retina so the use of the phrase “when HCs were optokinetically disabled” is not accurate and needs to be amended.
We agree, this was a mistake from our side. We corrected the text.
- Page 5 line 203: the discussion of the “subunit composition” of mGluR6 is confusing as this metabotropic receptor is a seven transmembrane domain protein without canonical “subunits”. Clarification of the same would benefit the reader.
We agree, thus we altered the text to avoid this confusion.
- The reference for Ganczer et al 2017 (page 12 line 254, page 13 line 485) is not included in the references section.
We agree and are also shocked that we left our own previous manuscript out from the reference list. We corrected the text and the reference list accordingly.
- Page 8 lines 309-311, the reference for mglur7 expression at BC terminals should not be Brandstatter et al 1997 (reference #91 detailing Kainate receptor distribution in the retina) but should instead be Brandstatter et al 1996 as this paper titled “Compartmental localization of a metabotropic glutamate receptor (mGluR7): two different active sites at a retinal synapse” shows data that suggests that mGluR7 is expressed at terminals of some cone bipolar cells.
We corrected this obvious mistake and replaced the reference with the correct one.
- Page 3 line 101: the phrase “where an S-ON/M-OFF” needs to be completed.
The sentence has been corrected.
- There are several typos in Figure 5: “ransient” or “rransient” instead of “transient” which need to be corrected.
Thank you for the warning, the typos now have been corrected.
- Legend of Figure 3 and 4: the photoreceptors have been denoted as “dark and light blue” but rod photoreceptors are depicted in grey.
Yes, we now included rod photoreceptors as well in the figure legend of the revision.
- page 5 line 179 kainate is misspelled as kainite and in line 206 the alpha appears as a typo.
The typos have been fixed.
- page 7 line 266: “accounted for the expression” should read “accounted for by the expression”.
The sentence has been altered accordingly.
- page 11 line 445: The phrase “both by and passive/active properties” needs to be amended.
The sentence has been altered accordingly.
- page 13 lines 519-521: the sentence starting “ It is known…” needs to be amended to clarify its meaning.
The sentence has been rephrased.
Reviewer 3 Report
The Authors propose an interesting study with the aim to review the recent knowledge regarding circuit elements of the mammalian retina that participate in shaping retinal ganglion cells response transience for optimal visual signaling. The manuscript is well written. Moreover in the revolutionary era of retinal gene therapy and retinal prosthetics it offers interesting insights and perspectives. Nevertheless small changes are required to improve the manuscript quality.
- In the Methods Section the Authors should specify the inclusion/exclusion criteria for the articles considered in the study.
- Are all the figures original?
- In the Conclusions Section the Authors should clarify possible future applications of these studies.
Author Response
We thank the reviewer for the critics and suggestions that made our manuscript even more thorough.
- In the Methods Section the Authors should specify the inclusion/exclusion criteria for the articles considered in the study.
Since this article is a review there is no Methods section. We intended to include all of those articles that provided significant data for the respective topic and the only exclusion criterion was irrelevancy. However, it may happen that contrary to our thorough literature review we overlook an important reference. Whenever our referees suggested to include some of these references (as it happened in the case of this manuscript as well) we are happy to complete the text and the reference list to make this review more comprehensive.
- Are all the figures original?
Yes, the schematics are all drawings of the first author created in Adobe Illustrator.
- In the Conclusions Section the Authors should clarify possible future applications of these studies.
We agree that it is important to emphasize possible applications of the topic thus we added a few extra lines on that in the conclusions section.